# The Modifying Role of Socioeconomic Position and Greenness on the Short-Term Effect of Heat and Air Pollution on Preterm Births in Rome, 2001–2013

**DOI:** 10.3390/ijerph16142497

**Published:** 2019-07-12

**Authors:** Federica Asta, Paola Michelozzi, Giulia Cesaroni, Manuela De Sario, Chiara Badaloni, Marina Davoli, Patrizia Schifano

**Affiliations:** Department of Epidemiology, Lazio Regional Health Service, ASL Roma 1, Via Cristoforo Colombo, 112-00147 Rome, Italy

**Keywords:** Socioeconomic factors, Urban environment, Green spaces, Preterm birth, Maternal exposure, Temperature, Air pollution

## Abstract

Urban green spaces have been associated with health benefits, but few studies have evaluated the role of greenness on pregnancy outcomes. We examined how the association between short-term exposure to heat and air pollution on the probability of preterm delivery is affected by the spatial variation of socioeconomic position (SEP) and greenness. We analyzed a cohort of newborns in Rome, from April to October of 2001–2013, defining preterm as births between the 22nd and the 36th week of gestation. We used a time series approach, with maximum apparent temperature (MAT), PM_10_, NO_2_, and O_3_ as exposure variables. As greenness indicators, we considered maternal residential proximity to green spaces and the Normalized Difference Vegetation Index (NDVI) within a 100 m buffer around each woman’s residential address. We enrolled 56,576 newborns (5.1% preterm). The effect of a 1 °C increase in temperature on the daily number of preterm births was higher in women with low SEP (+2.49% (90% CI: 1.29–3.71)) and among those living within 100 m from green spaces (+3.33% (90% CI: 1.82–4.87)). No effect modification was observed for NDVI or PM_10_. SEP was an important effect modifier of the heat-preterm birth relationship. The role of greenness in modifying this association between heat and preterm delivery should be further investigated.

## 1. Introduction

Preterm birth children, representing the 5–9% of the total births in Europe, are more likely to experience adverse health outcomes throughout childhood and into adulthood. The rates of preterm birth are increasing in many countries, reaching 11% [1,2,3,4]. The etiology of preterm birth is still not well known, but it is thought that it could be due to multi-factorial characteristics such as age, maternal conditions like preeclampsia or chronic infections, genetic predisposition, and socio-economic and lifestyle factors [5,6,7].

Climate change has been widely regarded as the greatest global threat for human health in the 21st century. As extreme meteorological events (e.g., heat waves, floods, droughts) are predicted to increase in frequency and magnitude in the future, it is important to identify the potential health effects of these meteorological conditions among vulnerable subpopulations such as pregnant women [8,9,10]. In recent years, the effect of high or low temperatures on pregnancy outcomes have represented an important research issue, and studies in different countries have shown a positive association between ambient temperature and several pregnancy outcomes, in particular between heat and preterm births [11,12]. Furthermore, evidence in the literature suggests that preterm births are associated with other urban environmental exposures such as air pollution [8,13,14,15,16,17]. Although biological plausibility is recognized, mechanisms linking air pollution to preterm births are still being investigated [14,18,19] and probably differ for each pollutant [20]. Several studies have reported different effects; the most consistent evidence on short-term air pollution effects is related to particulate matter with an aerodynamic diameter of 10μm or less (PM_10_), with effect estimates comparable to those of heat but more persistent and delayed in time [21,22]. 

In a previous study, we analyzed the effect of short-term exposure to temperature and air pollutants during warm and cold seasons on the probability of preterm births [22]. We found a significant association with heat and PM_10_ during the warm season. Furthermore, specific subgroups of women, with low education level and of young age, were more susceptible. However, other important risk factors were not analyzed. When evaluating the relationship between air pollution, ambient temperature, and preterm births, the potential role of different measures of socioeconomic position (SEP) should also be considered. It is well recognized that socioeconomic inequality plays a key role in pregnancy outcomes and that low socioeconomic groups can also be more exposed to environmental stressors and be more vulnerable [23,24,25,26]. Individual and area-based SEP could have independent associations with the susceptibility of individuals or could modify individual susceptibility to other risk factors [26,27]. In this context, SEP could also have a potential modifying effect on the association between ambient temperature, air pollution, and preterm births.

Green spaces can mitigate the urban heat island, absorbing solar radiation and cooling through evapotranspiration, and can create differential heat patterns within a city, especially at night [28]. However, the literature on this mitigation effect on pregnancy outcomes is still limited. It is reasonable to suppose that the distribution of green spaces within a city is differential by area-based SEP, hence it is important to study these two variables together. Considering the biological plausibility and the lack of epidemiological evidence on this topic [4], this study aimed to assess the association of maternal short-term exposure to heat and air pollution with the probability of preterm birth by analyzing whether spatial variation of area-based SEP and greenness could modify these relationships in the city of Rome.

## 2. Materials and Methods

### 2.1. Study Population

Rome covers an area of about 1290 km^2^, and most of the inhabitants are located within the Main Ring Road (Figure 1A). 

We previously collected data on all singleton live births in the city of Rome from 2001 and 2010, identified through the Certificate of Delivery Care Registry, which includes a wide range of information on maternal and fetal characteristics and clinical data on pregnancy and delivery. We followed the same inclusion criteria of our previous study [22], selecting pregnant women with singleton pregnancy that had a natural delivery or caesarean section with a spontaneous onset of labor that had reached at least 22 weeks of gestation. 

We extended the study period up to 2013 by selecting only births occurred in the warm season (1 April to 31 October). Then, we geocoded each woman’s residence at the time of delivery (using ArcGis 9.1 (Esri, Redlands, CA, USA)) focusing only on residents within the Main Ring Road.

### 2.2. Outcome Definition

We defined preterm birth as a live birth between the 22nd and the 36th week of gestation. We computed gestational age as the difference between the date of delivery and the first day of the last menstrual period.

### 2.3. Meteorological Variables and Air Pollution

We used daily maximum apparent temperature (MAT), which is an index of discomfort that includes both air and dew-point temperatures [29,30]. Apparent temperature was calculated for all three-hourly data measurements, and then the maximum daily value was calculated. We used temperatures measured at the airport station (Rome Ciampino) closest to the city center, and for every day, we assigned the same value of MAT to all mothers. 

We analyzed particulate matter with an aerodynamic diameter of 10 um (PM_10_), nitrogen dioxide (NO_2_), and ozone (O_3_) measured by the Lazio Environmental Protection Agency (http://www.arpalazio.gov.it/) at three monitoring sites (a background and two urban stations). We assigned daily values of PM_10_ (24-h mean), NO_2_ (24-h mean), and O_3_ (maximum eight-hour running mean) to all mothers. 

### 2.4. Greenness Indicators

We used two measures of green space in order to address different aspects, considering as buffer sizes those ones reported in literature on the beneficial effects of greenness [26,31,32]. We used the distance from a major green area [33], considering residential proximity as a surrogate for the access to green spaces. We computed the minimum straight line distance between home addresses and the boundary of the nearest major green space, identified from a shapefile (provided to us courtesy of Municipality of Rome) including all green urban areas, sports and leisure facilities and parks within the Main Ring Road (Figure 1B). Then we classified women as living within 100 m, between 100 m and 500 m, and beyond 500 m from green spaces [34].

We used the Normalized Difference Vegetation Index (NDVI) as a measure of surrounding greenness and a surrogate for general outdoor greenness [35]. NDVI ranges from −1 to 1 (with higher numbers indicating more greenness) is based on land surface reflectance of visible (red) and near infrared radiation [33,36,37,38,39]. We downloaded a Landsat 8-OLI/TIRS remote sensing image of NDVI from the USGS database (www.glovis.usgs.gov), with a spatial resolution of 30 m × 30 m (Figure 1C), for the 17/07/2015, assuming this image was representative of the overall follow-up period exposure [40]. We were confident that this assumption could not bias our greenness indicator since Orioli et al. demonstrated that the correlations between NDVI summer values across the period from 2001 to 2013 plus 2015 were always very high (about 0.80), denoting no substantially changes in the amount of greenness in the city of Rome [40]. We selected a summer day, chosen as it had the least amount of cloud cover at the time of year when leaf development would be at a maximum. Finally, we calculated the average of NDVI in each 100 m buffer around of the mothers’ residential addresses and classified each woman using NDVI’s tertiles [34].

### 2.5. Socioeconomic Position

We considered a small area-level indicator of socioeconomic position (SEP): the index is based on data from the 2001 census in Rome, derived from a factor analysis that included education, occupation, home ownership, family composition, crowding, and immigration at census block level (500 inhabitants per block, on average) [41]. We categorized the original quintiles of this index in three levels: low (fourth and fifth quintiles), medium (third quintile) and high (first and second quintiles) (Figure 1D).

### 2.6. Statistical Analysis

We used Poisson generalized additive models to check the linearity of the relationships and to estimate the associations between environmental exposures and the daily proportion of preterm births. We included the daily number of pregnancies at risk as offset in the models. We adjusted the models for long-term trend and seasonality (considering natural splines with degrees of freedom equal to number of years minus one for the long-term and with degrees of freedom equal to number of months minus one for seasonality), and holidays (a categorical variable which assumes value “1” in national or local holiday days and “0” otherwise). 

We first studied MAT and each pollutant separately, and we considered the adjusted models only when collinearity did not arise. We checked for collinearity among the exposures using the variance inflation factor (VIF) with values of VIF > 10 indicating presence of collinearity. 

In our previous study, we studied the lag structure of the relationships with a distributed lag model (DLM), allowing a lag structure of up to 30 days [22]. In accordance to the previous study, we considered a lag of 0–2 days for MAT, NO_2_, and O_3_, while a lag of 12–22 days before delivery was considered for PM_10_. Each exposure was included in the model as a linear term at the chosen lag. 

To identify whether greenness and small-area SEP modified the effect of MAT and air pollutants on the probability of preterm birth, we included an interaction term in the model between each potential effect modifier and the environmental indicators, considering a *p*-value less than 0.10 as threshold. We also stratified by each potential effect modifier considered in this study (green indicators and SEP) and maternal age and education. 

We used SAS 9.2 (SAS Institute Inc., Cary, NC, USA) and ArcGis 9.1 (ESRI, Redlands, California, USA) to create datasets while we carried out the analyses with R software (R Foundation for Statistical Computing, Vienna, Austria) (www.r-project.org).

## 3. Results

We considered 56,576 singleton live births during the study period, from mothers delivering in Rome and residing within the Main Ring Road; 2910 (5.1%) were preterm. The maternal residential address was available for all women included in the study. In our cohort, 28,946 (51%) mothers lived within 100 to 500 m from a green space with 5.2% preterm births, while only 16% (9013) lived within 100 m of a green area and had a 5.4% of preterm births. No relevant differences in preterm birth distributions were found by NDVI (Table 1). 

As shown in Table 1, while the distribution of maternal age and education level was similar across the different green space categories, a high SEP was more frequent among women living within 100 m and within 100–500 m from green spaces (43.5% and 46.2%, respectively) compared to those living beyond 500 m (34.8%). We found a similar trend when considering the tertiles of NDVI: a high SEP was more frequent among mothers living in areas with NDVI > 0.36 (48.0%), compared to the two lowest categories (38.4% and 40.0%, respectively).

The percentile distribution of daily MAT and air pollutants between 1st April and 31st October 2001–2013 is reported in Table 2. The MAT interquartile range (IQR) was 10.3 °C and the O_3_ showed the highest variability among the pollutants (Table 2). We observed moderate correlations between MAT and O_3_ (ρ = 0.6) and between PM_10_ and NO_2_ (ρ = 0.5) (Appendix A).

The overall model showed that MAT was associated with an increased probability of preterm birth; as shown in Table 3, we observed a percent change of 2.00 (90% CI: 0.98–3.03) in the proportion of preterm births per 1 °C increase in MAT with a lag of 0–2 days. Regarding air pollutants, only PM_10_ was associated with an increased probability of preterm birth; for an IQR increase (11 μg/m^3^) in PM_10_ with a lag of 12–22 days before the delivery, the percent change in the proportion of preterm births was equal to 8.16 (90% CI: 2.38–14.26). For NO_2_ and O_3_, no significant effect on preterm births was observed (Table 3). No collinearity was observed among exposures.

Socioeconomic position and residential proximity to green spaces modified the effect of temperature in our analysis. We found a percent change of 3.33 (90% CI: 1.82–4.87) in the proportion of preterm births per 1 °C increase in MAT in women living within 100 m from green areas, while those living further away from green spaces (>500 m) showed a significantly lower effect of temperature on preterm births of 1.03% (90% CI: −0.19–2.28) compared to women living within 100 m (reference group) (Figure 2). 

For SEP, women with low SEP were more susceptible to the effect of temperature on the probability of preterm birth (2.49% increase per 1 °C increase in MAT (90% CI: 1.29–3.71) than women with high SEP (1.01% increase per 1 °C increase in MAT (90% CI: −0.15–2.18)) (Figure 2). No effect modification by residential surrounding greenness was found.

Furthermore, none of the considered factors modified the effect of PM_10_ on the preterm birth probability (Figure 3). However, results suggested an increasing trend in PM_10_ effects with an increasing distance from green areas.

When we included a double interaction between residential proximity to green spaces and SEP in the model, the highest effect of temperature on the preterm birth probability was among women with low SEP and living within 100 m from a major green space. In this group, we detected a percent change of 5.28 (90% CI: 2.97–7.65; *p*-value of interaction term = 0.048) per 1 °C increase in MAT compared to women with high SEP living within 100 m (+2.2%; 90% CI: 0.2–4.2) (Appendix A).

Considering the double interaction between residential proximity to green spaces and maternal age, we found a 4.4% increase (90% CI: 2.47–6.42) in the proportion of preterm births among women aged 30–36 years and living within 100 m from green areas for a 1 °C increase in MAT vs. a 1.3% increase (90% CI: −1.23–3.90) in women aged 37 and over living within 100 m (*p*-value of interaction term = 0.082; Appendix A). No effect modification was found when we studied the double interaction between residential proximity to green spaces and education level.

## 4. Discussion

This study is one of the first to evaluate the association of maternal exposure to high ambient temperature and urban air pollutants with the probability of preterm birth by analyzing the spatial variation in SEP and green spaces as effect modifier. Although we considered a longer time series adding three years to previous analysis [22], our results confirmed that, in Rome maternal exposure to heat and PM_10_ was associated with a short-term increase in the probability of preterm birth during the warm season. A 1 °C increase in MAT was associated with a percent change of two in the daily proportion of preterm birth, consistent with our and other previous studies [10,11,22]. As for air pollutants, we found a significant effect of exposure to PM_10_ in the second and third week before delivery with an 8% increase in the daily proportion of preterm birth for an IQR increase of PM_10_ (11 μg/m^3^). 

The effects of heat and air pollution on preterm births have been discussed in several studies and reviews [4,8,11,16,17,20,22]. In particular, the short-term effect of high temperature on pregnant women could be explained by several biological mechanisms: the secretion of oxytocin and prostaglandin (PGF2α) has been shown to increase under heat stress conditions and to induce labor [42,43]. In addition, the onset of labor could be induced by heat-related dehydration, reducing maternal fluid level, and consequently the fetus blood volume [44]. The underling biological mechanisms of the short-term effect of air pollution on preterm births were less clear [14], and seemed to involve both endothelial dysfunction and oxidative stress leading to placental inflammation [19,45].

Less is known on the individual characteristics that may worsen or mitigate these effects. Even if our study adopted a time series design, we were able to analyze socioeconomic position and closeness to major green areas at residential address of each mother as potential effect modifiers, considering residency from population registry data. We found an interaction between residential proximity to a major green space and temperature: women living within 100 m from a green area had a greater probability of preterm birth for every 1 °C increase in temperature compared to women living beyond 500 m. No effect modification was found when considering NDVI within 100 m from each residential address. Although there is a growing literature on the beneficial health effects of urban green spaces, existing evidence on pregnancy outcomes and surrounding greenness is still inconsistent [34,35,46,47]. In Rome, contrary to expectations, closeness to green spaces was associated with a higher probability of preterm birth. Our study did not seem to support the hypothesis of potential cooling effect due to greenness, but this could be due to the use of the same temperature level for all women. Furthermore, it should be taken into account that mitigation of temperatures and of the urban heat island occurs mostly at night, when women are indoor, with closed windows and air conditioning. All of these aspects could have impacted our results. The lack of literature on this topic limits comparison with other studies considering both meteorological parameters and the distance at which the mitigation is still effective [48,49]. A recent study focusing on heat-related mortality, conducted in Vietnam, highlighted the difficulty to choose the appropriate meteorological parameters when dealing with the impact of greenness on human health, given the complex interaction between vegetation and air temperature at urban levels [48]. Another aspect to consider when interpreting our findings could be the potential heterogeneous utilization patterns of the nearby green space between women living very close to a green area (less than 100 m) and those living farther. It could be that the former group, given the closeness to green spaces, might spend more time outdoors, thus increasing the level of heat exposure, compared to the latter. It is also possible that closeness to green spaces might be a proxy of other life styles variables we were not able to consider. 

Our study suggested a potential mitigating effect of residential proximity to green spaces on PM_10_-preterm birth association. This is coherent with literature showing that vegetation could absorb particles decreasing the levels of air pollution, and with studies showing a lower traffic density nearby green areas [28]. We could not compare this finding with other studies because, to our knowledge, this was the first study to evaluate the role of greenness as modifier of temperature/pollution and preterm births relationship. 

In our study the association between MAT and preterm birth was also modified by SEP with a higher effect in women with low SEP, equal to about 3% in the daily proportion of preterm birth for every 1 °C increase in MAT. A similar increased probability was found in women with primary school vs degree in our previous study in which we considered maternal education [22]. Although the results were quite similar, we believed that SEP as a composite index with its multidimensionality, could give us more comprehensive view on this aspect, compared to education level that represents only one dimension [26,27]. One explanation could be due to the potential worse health status of low SEP people due to limited access to health care and living in areas with poor housing and worse environmental conditions [26,50]. In particular, we identified a subgroup of women highly susceptible to the heat effects: those with low SEP and living very close to green areas (within 100 m) showed an increase of about 5% in the daily proportion of preterm births for every 1 °C increase in MAT. The literature suggests that subjects with low SEP spent more time in their local area, hence the time to local exposure was greater and they might use local green spaces close to their residence more [51]. Many other risk factors, not directly measured in this study, might be reasonably associated with a low SEP, such as lack of air conditioning, poor diet, and smoking during pregnancy, that should be considered [52]. Since our study was based on administrative databases, which represent a point of strength in terms of sample size, cost, and feasibility, the opportunity to consider other potential individual risk factors was limited.

Some limitations of this study have to be discussed. Since there was no universally accepted definition of urban green space, we decided to use two indicators of greenness—one derived from satellite data (NDVI) and one based on the distance between residential address and the nearest major green area provided by Municipality of Rome. For NDVI, we considered a single satellite image taken at a single day in July. Although we did not account for the potential temporal variation in greenness, a previous study conducted in Rome showed negligible changes in the quantity of greenness during the study period [40]. Neither of the chosen indicators allowed us to distinguish between different types of vegetation, vegetation volumes, and types of green space (street trees or gardens). The choice of buffer’s width around woman address of residence was another critical point: since we were interested to catch the cooling effect of greenness, we analyzed both green indicators within a small buffer size as suggested by literature [31,32,49]. Furthermore, we assumed that all people have equal access to the green space closer to the ward of residence, but this might not be true [53]. All of these features might play an important role in temperature mitigation, and they need to be accounted for in this kind of analysis [28,37].

Also, we used an indirect assessment of exposure: meteorological data were retrieved from a fixed monitoring weather station located in Rome (Ciampino Airport), and pollutants were derived from three fixed monitoring stations within the city. Therefore, an average measure of outdoor temperature and of air pollutants for all women residing in the municipality of Rome, within the Main Ring Road, was used. This could have induced non-differential misclassification of exposure because it was unrelated to the outcome, thus biasing the estimated effects toward the null value. Furthermore, it is important to take into account that personal exposure may be modified by the amount of time spent indoor, by the use of air conditioning, and by residential mobility, which we were, at this stage, unable to consider. Moreover, both effect modifiers (SEP and greenness) might have been affected by a potential misclassification due to the implicit assumption of residential stability during pregnancy. Our research relied on maternal residential addresses at the time of delivery, and we were not able to reconstruct residential mobility during pregnancy.

Finally, the different spatial resolutions between the main exposures (measured at the city scale) and the effect modifiers (measured at the individual scale) could have probably inflated the bias concerning the studied interactions. A major improvement in the analysis could be the use of the individual exposure levels through spatio-temporal models for both temperature and air pollutants to catch the true role of green areas and SEP as effect modifier.

## 5. Conclusions

The study confirms that maternal exposure to heat and PM_10_ is associated with a short-term increase in the preterm-birth probability. This association appears to be modified by socioeconomic position and closeness to green spaces. Contrary to expectations, living close to green areas seems to increase the preterm birth probability when exposed to heat, especially in women of low socioeconomic position. 

It is interesting to note that, while the existing evidence confirms a general salutary impact of greenness on health, much remains to be clarified on the particular pathways involved and, above all how these may change by social context, subgroups, and pregnancy outcomes. For all these reasons, the role of greenness as potential modifier of hazardous exposures in urban areas should be further investigated.

## Figures and Tables

**Figure 1 ijerph-16-02497-f001:**
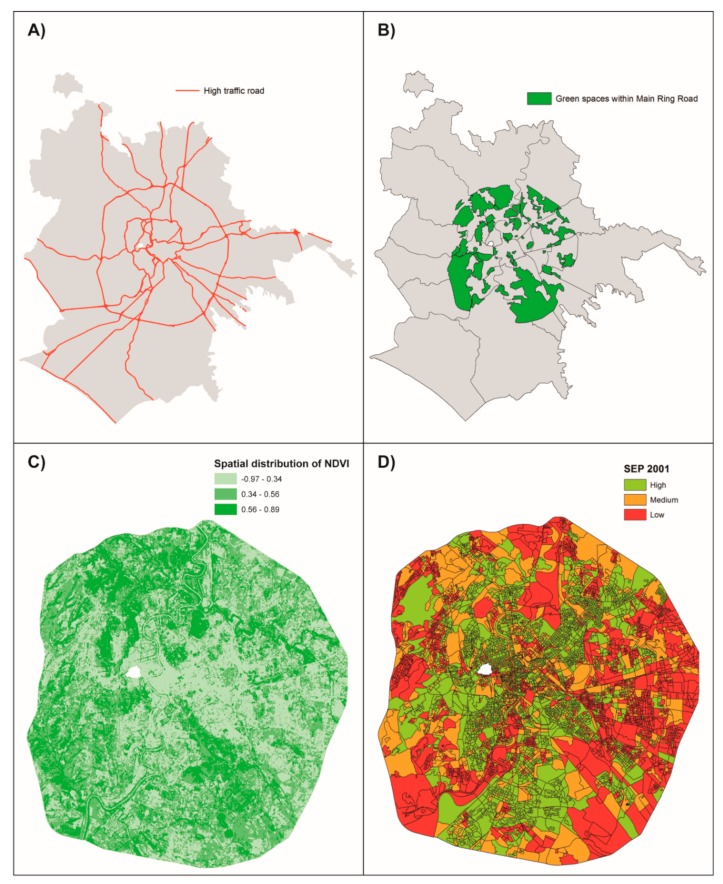
Distribution of greenness and socioeconomic position within the Main Ring Road in Rome. (**A**) Map of Rome with high traffic roads; (**B**) Major green space within the Main Ring Road; (**C**) Spatial distribution of the Normalized Difference Vegetation Index (NDVI) within the Main Ring Road according to tertiles; (**D**) Map of Rome within the Main Ring Road by socioeconomic position (SEP).

**Figure 2 ijerph-16-02497-f002:**
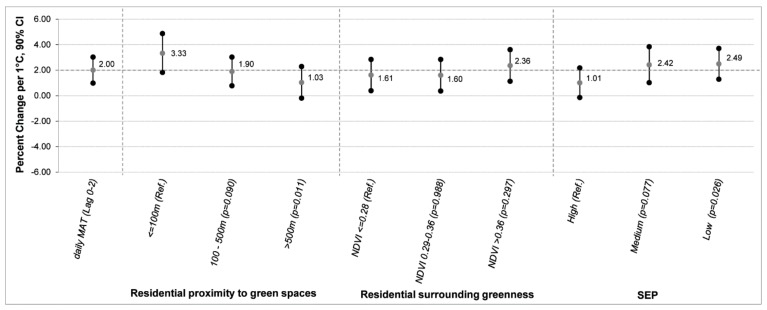
Percent change * in daily preterm births by 1 °C increase of MAT (Lag 0–2). * Effect estimates derived by independent Poisson regression models including one covariate per time. Ref.: reference category. p: *p*-Value of interaction terms of each modality compared with the reference category.

**Figure 3 ijerph-16-02497-f003:**
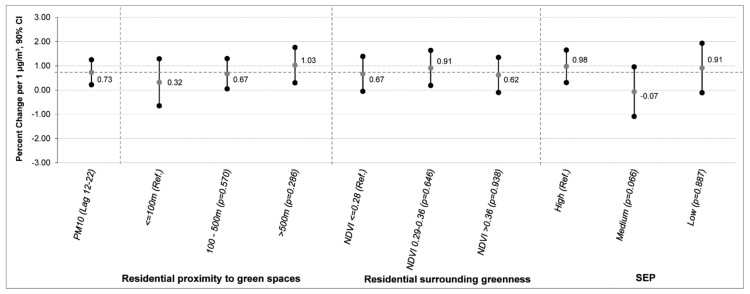
Percent change * in daily preterm births by 1 μg/m^3^ increase of PM10 (Lag 12–22). * Effect estimates derived by independent Poisson regression models including one covariate per time. Ref.: reference category. P: p-Value of interaction terms of each modality compared with the reference category.

**Table 1 ijerph-16-02497-t001:** Characteristics of the birth cohort in the metropolitan area within the Main Ring Road (Rome).

Variable	Entire Cohort *n* (%)	Residential Proximity to Green Spaces			Residential Surrounding Greenness		
		≤100 m *n* (%)	100–500 m *n* (%)	>500 m *n* (%)	NDVI T1 (≤0.28) *n* (%)	NDVI T2 (0.29–0.36) *n* (%)	NDVI T3 (>0.36) *n* (%)
Study population	56,576	9013 (15.9)	28,946 (51.2)	18,617 (32.9)	19,115 (33.8)	19,239 (34.0)	18,222 (32.2)
Preterm birth	2910 (5.1)	487 (5.4)	1491 (5.2)	932 (5.0)	989 (5.2)	944 (4.9)	977 (5.1)
Maternal Age (years)							
<30	13,644 (24.1)	2156 (23.9)	6850 (23.7)	4638 (24.9)	4590 (24.0)	4608 (24.0)	4446 (24.4)
30–36	30,485 (53.9)	4846 (53.8)	15,707 (54.3)	9932 (53.4)	10,315 (54.0)	10,443 (54.3)	9727 (53.4)
≥37	12,447 (22.0)	2011 (22.3)	6389 (22.1)	4047 (21.7)	4210 (22.0)	4188 (21.8)	4049 (22.2)
Education Level							
Primary school	18,660 (33.0)	2899 (32.2)	9426 (32.6)	6335 (34.0)	6429 (33.6)	6343 (33.0)	5888 (32.3)
High school	24,268 (42.9)	3936 (43.7)	12,329 (42.6)	8003 (43.0)	8022 (42.0)	8415 (43.7)	7831 (43.0)
Degree	13,648 (24.1)	2178 (24.2)	7191 (24.8)	4279 (23.0)	4664 (24.4)	4481 (23.3)	4503 (24.7)
Socioeconomic position							
High	23,795 (42.1)	3924 (43.5)	13,384 (46.2)	6484 (34.8)	7347 (38.4)	7704 (40.0)	8744 (48.0)
Medium	12,349 (21.8)	1985 (22.0)	5972 (20.6)	4392 (23.6)	4862 (25.4)	3804 (19.8)	3683 (20.2)
Low	20,432 (36.1)	3104 (34.4)	9590 (33.1)	7738 (41.6)	6906 (36.1)	7731 (40.2)	5795 (31.8)

**Table 2 ijerph-16-02497-t002:** Description of environmental exposures during the period April–October in Rome, 2001–2013.

Exposure	Min	25° pctl	50° pctl	75° pctl	Max
Warm Season (April–October)					
Meteorological					
daily maximum apparent temperature (°C)	4.1	20.6	25.8	30.9	39.7
Air pollution					
PM_10_ (μg/m^3^)	7.0	24.0	30.3	38.0	181.7
Ozone (μg/m^3^)	7.7	78.6	94.8	111.4	199.2
NO_2_ (μg/m^3^)	11.9	42.1	53.0	63.5	110.1

pctl = percentile.

**Table 3 ijerph-16-02497-t003:** Percent change in daily number of preterm by temperature and air pollutants. Rome, 2001–2013.

Environmental Exposures ^a^	% Change per Unit Increase (90% CI)	% Change per IQR (90% CI)	IQR
Warm Season (April–October)			
MAT (Daily maximum apparent temperature) (Lag 0–2) ^b^	2.00 (0.98, 3.03)	22.74 (10.59, 36.21)	10.3
Air pollutants ^c^			
PM_10_ (μg/m^3^) (Lag 12–22)	0.73 (0.22, 1.25)	8.16 (2.38, 14.26)	10.7
Ozone (μg/m^3^) (Lag 0–2)	0.03 (−0.19, 0.25)	0.53 (−3.40, 4.61)	30.4
NO_2_ (μg/m^3^) (Lag 0–2)	0.20 (−0.14, 0.55)	6.39 (−4.27, 18.22)	18.2

a Each model was adjusted for long term trend, seasonality and holidays. b Percent change from model adjusted for PM10 (lag 12–22). c Percent changes derived from independent models including one pollutant per time, adjusted for MAT (lag 0–2). IQR = interquartile range.

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
