# Peer review of "The Modifying Role of Socioeconomic Position and Greenness on the Short-Term Effect of Heat and Air Pollution on Preterm Births in Rome, 2001–2013"

_ijerph, 2019, doi:10.3390/ijerph16142497_

Round 1

Reviewer 1 Report

This study assessed the effects of socioeconomic position (SEP) and greenness on the association of short-term exposure to heat and air pollution with the risk of preterm delivery. They found women of a lower socioeconomic position living close to green areas have a higher risk of preterm birth when exposed to heat. This is an interesting study, but I still have some comments:

1. There were several grammar mistakes, even in the abstract. The English language need to be proved.

2. Line 102-103, it is not clear how the authors calculate the air pollution levels for each participant. Please be more specific.

3. From the maps, it seems the there is a colinearity between SEP and greenness. The authors are suggested to assess it. 

Author Response

1. There were several grammar mistakes, even in the abstract. The English language need to be proved.

v  We checked our manuscript by a native English speaking colleague and change it accordingly.

2. Line 102-103, it is not clear how the authors calculate the air pollution levels for each participant. Please be more specific.

v  We assigned the same daily 24-h mean for PM10 and NO2 while the daily maximum 8-h running mean for O3 to the addresses of all mothers residing in the city. All women had the same daily values for these three different air pollutants. According to the reviewer suggestion, we modified slightly the text, lines 100-104, to better clarify the attribution of pollutants exposure.

3. From the maps, it seems the there is a colinearity between SEP and greenness. The authors are suggested to assess it. 

v  If the reviewer is referred to SEP and NDVI, we never considered these two exposures together in the models and for this reason we did not take into account collinearity issue. Moreover since they were categorical variables, we checked the concordance between these variables with Cohen's kappa coefficient (K) and we found a value equals to -0.05 indicating no agreement. We checked also the agreement between SEP and Residential proximity to green spaces and we found a K equals to 0.02. These low levels of agreement could be due to a different spatial resolution between SEP and the two greenness indicators that we used.

Reviewer 2 Report

Overall this was a really interesting paper that adds to the literature on this growing field. A few major and minor comments. 

Major comments: 

The authors used a Landsat 8 image from 2015 but their time period is 2001-2013. Orioli et al., the reference used as the assumption that the image is representative of the follow-up period, also uses a Landsat 8 image outside of the follow-up with no justification that it is representative of the study period. The authors either need to use a Landsat 5 image from during the follow-up (since Landsat 8 was not launched until 2013) or find a reference to back up the claim that there was no or very little change in greenness over that time-period. It is possible that there was no development in central Rome during that time but more information on the topic is warranted. 

A VIF of 10 is very high. Please provide justification why that was chosen and not something lower. 

The authors switch between risk, rate, and percent change throughout. Percent change does not seem like the most easily understood outcome measure. If the authors keep the description as rate/day, please consider using births/day as the outcome measure or something more consistent. 

A few of the conclusions overstate the results a bit. The suggestion that poorer women are not aware of the risks seems more like a hypothesis for earlier in the discussion. This paper also did not seem to find a benefit of proximity to greenspace so the salutary impact does not seem to be confirmed here. Otherwise, the rest of the conclusions are very interesting. 

Minor comments: 

"small area" SEP is not generally understood so the authors should avoid using that in the introduction and use it in the methods after it is defined. 

Figure 1C. says the average NDVI is in the 100m buffer but it's just the average NDVI throughout the area I believe. 

Please explain the significance of "days of holiday" in the adjusted model. 

Author Response

Major comments: 

The authors used a Landsat 8 image from 2015 but their time period is 2001-2013. Orioli et al., the reference used as the assumption that the image is representative of the follow-up period, also uses a Landsat 8 image outside of the follow-up with no justification that it is representative of the study period. The authors either need to use a Landsat 5 image from during the follow-up (since Landsat 8 was not launched until 2013) or find a reference to back up the claim that there was no or very little change in greenness over that time-period. It is possible that there was no development in central Rome during that time but more information on the topic is warranted. 

v  Thanks to discussing this critical point, but maybe there was a misunderstanding on the reference that we used to justify our choice. In our study we had surely the limit that we assessed greenness based on a remote sensing image taken at a given moment and this did not allow to address temporal variation. We cited Orioli et al. because they made a sensitivity analysis focused on the assumption that this single image could be representative of the entire study period in terms of no substantially changes in the amount of greenness in the city of Rome. As Orioli and colleagues reported in the supplementary material (https://doi.org/10.1289/EHP2854), the correlations between NDVI summer values across all the years were always very high (about 0.80); based on these results and aware that our study area is even more limited, since we considered only the area within the Main Ring Road, we were confident that using this single image from Landsat 8 could not bias our greenness indicator. We added a sentence on this limit in the discussion, lines 287-290.

A VIF of 10 is very high. Please provide justification why that was chosen and not something lower. 

v  Moderate multicollinearity is fairly common since any correlation among the independent variables is an indication of collinearity, for this reason we were interested to detect severe multicollinearity and fixed a threshold equals to 10 following a rule of thumb suggested by literature (Kutner, M. H.; Nachtsheim, C. J.; Neter, J. (2004). Applied Linear Regression Models (4th ed.). McGraw-Hill Irwin.) and also by the help of the Stata command  that we used to check collinearity.

The authors switch between risk, rate, and percent change throughout. Percent change does not seem like the most easily understood outcome measure. If the authors keep the description as rate/day, please consider using births/day as the outcome measure or something more consistent. 

v  Just to clarify, we used percent change to quantify the association between exposure and outcome; while “risk” or “rate” were used as measure of outcome. Moreover, preterm births were a proportion, so we decided to standardize using the wording “preterm birth probability” or “proportion of preterm births” throughout the text.

A few of the conclusions overstate the results a bit. The suggestion that poorer women are not aware of the risks seems more like a hypothesis for earlier in the discussion. This paper also did not seem to find a benefit of proximity to green space so the salutary impact does not seem to be confirmed here. Otherwise, the rest of the conclusions are very interesting. 

v  Thanks for the suggestion, we modified the first part of the conclusion, lines 316-320, deleting the hypothesis and strengthening the main results about of effect modification.

Minor comments: 

"small area" SEP is not generally understood so the authors should avoid using that in the introduction and use it in the methods after it is defined. 

v  We substituted "small-area" with “area-based” in the introduction because for us is important to distinguish it from the individual one.

Figure 1C. says the average NDVI is in the 100m buffer but it's just the average NDVI throughout the area I believe. 

v  Yes you are right, there was a mistake in the title, but what we reported in the footnote is correct. Figure 1C is the NDVI with a spatial resolution of 30m x 30m. We changed the title and the legend accordingly.

Please explain the significance of "days of holiday" in the adjusted model. 

v  With "days of holiday" we included in the model a categorical variable with two modalities 0 or 1; it assumes value “1” in those days when in Italy it is a national and local holiday (for example Easter, 15th of August, etc.) and “0” otherwise. We add a sentence in the text, lines 137-138.

Round 2

Reviewer 2 Report

Thank you for the thorough responses to my comments. I have one final minor comment. 

In regards to the Landsat image used for NDVI calculation and the reference used: After being directed to the supplemental materials section of the Orioli et al. paper, it is clear that the authors justified their use of the single image. However, the layout of the Orioli et al. paper and the inclusion of that fairly major justification in a separate section than the measurement of greenness makes it difficult to find for the reader. In addition to the helpful addition to the discussion, I think it would make it clearer to quickly summarize the results of Orioli's supplemental analysis in this methods section than to rely on readers of this manuscript being able to find the hidden analysis in that paper. 

Author Response

In regards to the Landsat image used for NDVI calculation and the reference used: After being directed to the supplemental materials section of the Orioli et al. paper, it is clear that the authors justified their use of the single image. However, the layout of the Orioli et al. paper and the inclusion of that fairly major justification in a separate section than the measurement of greenness makes it difficult to find for the reader. In addition to the helpful addition to the discussion, I think it would make it clearer to quickly summarize the results of Orioli's supplemental analysis in this methods section than to rely on readers of this manuscript being able to find the hidden analysis in that paper. 

v  We add a sentence in the methods section as required, lines 121-124. We highlight changes in blue to distinguish from the previous variations (in yellow).

v  Only to clarify, we checked our manuscript by a native English speaking colleague since the first revised version. Moreover, we read again to correct some minimal typos and, also in this case, we highlight these changes in blue.